# The eco-evolutionary assembly of complex communities with multiple interaction types

Gui Araujo & Miguel Lurgi ⬤ ✉

Identifying the mechanisms that generate structure in complex ecological communities is fundamental for understanding their assembly. Yet a comprehensive picture of how ecology and evolution combine to generate these patterns remains limited. We use an eco-evolutionary model of community assembly that incorporates interaction-driven population dynamics and evolutionary processes, including speciation and inheritance of interactions, to unveil the mechanisms generating and maintaining biodiversity in complex species interaction networks. Importantly, our model unpicks the effects of selection of interaction types from those of inheritance by comparing evolutionary assembly with invasion-based assembly under different combinations of interaction types. We find that a cost-benefit balance in accumulating interactions separates communities into two distinct types. Weakly beneficial interactions produce sparse, competition-dominated networks, whereas strongly beneficial interactions generate highly mutualistic, more connected communities. Mutualism, driven by both selection and inheritance, facilitates the emergence of large communities with increased complexity. Comparing model results with empirical patterns from microbial communities, we identify potential drivers of ecosystem assembly and characteristic interaction structures. Our results provide a classification system of complex ecosystems based on their composition of ecological interactions, thus generating testable hypotheses on the conditions under which different community types (mutualistic vs. competitive) might emerge.

Unveiling and understanding the processes that generate and maintain biodiversity is a central focus of ecological research. Both evolutionary (e.g. speciation) and ecological (e.g. invasions) assembly mechanisms, and how they are shaped by biotic and abiotic factors such as species interactions and environmental conditions, have been shown to promote the complexity observed in natural communities[1–6]. Clarifying the relative importance of different mechanisms that foster biodiversity, while maintaining the stability of complex communities thus assembled, is fundamental to understand ecosystems[7,8] and to inform future conservation strategies[9,10].

Five decades ago, Robert May[11] demonstrated that ecological systems with a large number of species and high connectivity are less likely to be stable than simpler ones. This seminal finding sparked a decades-long search for the organisational features that enable natural communities to increase in complexity without compromising stability or persistence. May's result referred to randomly connected communities (i.e., not subjected to a deterministic structure or assembly processes) represented by randomly sampled community matrices. Subsequent work has shown that particular network structures can alter the complexity-stability relationship[12], and that in empirical systems a large number of species often translates into low connectivity between them[3]. Furthermore, beyond structural patterns, the relative composition of interaction types, such as mutualism, competition, and consumer-resource interactions, has emerged as a

Department of Biosciences, Swansea University, Singleton Park, Swansea SA2 8PP, UK. ✉e-mail: miguel.lurgi@swansea.ac.uk

key factor influencing the outcomes of community assembly and the stability of the resulting communities[13–17]. Only recently has theoretical research begun to explore how the interplay between different interaction types shapes the temporal assembly of complex communities, with recent studies revealing the key role of mutualistic interactions in fostering diversity (e.g.,[18]). The next challenge is to uncover the mechanisms driving the emergence of different compositions of interaction types, the structural patterns of the resulting species interaction networks, and the influence of these patterns and processes on the diversity-connectivity relationships that underpin the complexity of ecological communities.

Evolutionarily, speciation is a major driver of community assembly. Speciation, acting on an existing pool of species and traits, together with subsequent selection, has been shown to recover patterns of biodiversity and network structure observed in complex model ecosystems[4,19]. These findings suggest that speciation-driven assembly offers a key pathway for the emergence of complexity in ecological communities. Comparing speciation-driven evolutionary assembly with ecological assembly via repeated invasions helps clarify the distinct contributions of evolutionary processes to community structure[20]. Considering both speciation and invasion processes together, and their interplay, enables the investigation of two key mechanisms shaping community structure: selection of interaction types and inheritance of interactions. To date, theoretical studies of evolutionary community assembly have primarily focused on single-type species interaction networks such as food webs or mutualistic networks[4,5,19–25]. The combined effects of multiple interaction types have so far only been explored in a preliminary manner in evolutionary models of community assembly, with an emphasis on the study of stability[26], leaving their broader role in shaping ecological complexity largely unresolved.

Network structure not only determines biodiversity, complexity, and stability but can also characterise distinct types of ecological communities. These structural properties often emerge from different ecological and evolutionary assembly mechanisms, as observed across diverse ecosystems[27]. For instance, in microbial communities associated to phytoplankton cells in the phycosphere, community composition is shaped by host identity and metabolic profile[28,29]. More generally, across host-associated microbiomes, microbial succession is thought to be driven by cross-feeding interactions between microbes and with the host[30]. The discovery of universal macroecological patterns of abundance and variability in microbial communities has motivated efforts to identify general network structures that can give rise to such patterns[31,32]. Non-interactive population dynamic models have successfully reproduced many of these patterns and have been proposed as null models for microbial community structure[31,33,34]. However, such models fail to capture non-random correlations in species abundances observed in these communities, a caveat that has been addressed by adding sparse and weak interactions between community members[35,36]. Together, these findings highlight the critical role of the structure of ecological interactions in shaping microbial communities and the need to identify the assembly mechanisms that give rise to such structural organisation. Similar challenges are observed across ecosystems types and scales. For example, in mutualistic networks comprising interactions between plants and their animal pollinators, characteristic patterns of network modularity and nestedness have been suggested to arise through adaptation and niche optimisation[25] or speciation[4]. A holistic eco-evolutionary understanding of how these processes and their interplay determine community assembly can shed further light on ecological structure across scales.

Here, we develop a deeper mechanistic understanding of community assembly by investigating the evolutionary emergence of complex ecological networks involving multiple interaction types. We use a dynamical model that we previously developed[26] in which

mutualistic, consumer-resource, and competitive interactions shape the outcomes of speciation and invasion events driving community assembly and complexity. Interactions between species determine how they influence one another, potentially triggering extinction cascades. Over time, successive assembly events and ecological filtering give rise to stable interaction network structures. We find that the strength of ecological interaction benefits defines a threshold that separates two distinct community types: one dominated by competition, and the other by mutualism. Moreover, mutualism and speciation jointly promote increasing network complexity and species diversification. Nonetheless, species richness can increase independently of mutualism and speciation if enough invasion events occur and network connectivity is low. By analysing longitudinal data from microbial communities[31,35], we link macroecological patterns to underlying assembly mechanisms. We hypothesise that strong benefits from ecological interactions can favour the emergence of highly interactive microbial communities that display patterns of abundance correlations characteristic of these systems. By linking eco-evolutionary assembly mechanisms to emergent community structure and observed macroecological features, our results advance understanding of the processes that shape complexity in natural communities.

## Results

We analyse a step-wise community assembly model in which new species are introduced sequentially to the existing community via two distinct mechanisms: (1) evolutionary speciation and (2) invasion. We start the assembly process with small communities comprising five non-interacting species.

Assembly by evolutionary speciation: In this scenario, new species arise as a variation of a randomly selected parent species and inherit their ecological interactions with limited fidelity. Inheritance with changes in species interactions represents mutations in foraging, morphological, or physiological traits that biologically determine interactions with other species (e.g., feeding modes, body morphology, metabolic pathways, signaling mechanisms, etc.). A parameter $\Delta$ defines the maximum number of interactions by which the offspring may differ from the parent species. Smaller values of $\Delta$ correspond to more constrained inheritance (i.e., greater similarity to the parent), a regime we term strong inheritance. For each speciation event, all interactions of parent species are copied into the offspring, although not faithfully: an integer $d$, representing the total number of interaction changes, is then drawn uniformly from the interval $[1, \Delta]$. Interactions are randomly removed and/or added from the offspring until the total number of modifications, relative to the parent, equals $d$. For a given $d$, the number of interactions created is uniformly chosen from $[1, d]$ and the number of interactions destroyed is $d$ minus that number. If there are not enough interactions inherited (or no interactions, as in the initial state), then new interactions are created whenever an inexistent one would be destroyed (and vice-versa). Speciation is thus a flexible process, with the average degree of inheritance controlled by $\Delta$: larger values lead to weaker average inheritance. If a parent species has no interactions (i.e., an isolated species), at least one interaction must be added ($d \geq 1$) to ensure variability.

Assembly by invasion: We then define a purely ecological scenario in which new species are introduced without any inherited interactions. Instead, each invader is assigned a connectivity value drawn uniformly from the interval $[\rho_1, \rho_2]$. This value defines the probability of forming an interaction with each existing species in the community. As invaders differ in their connectivity, community-level connectivity can shift over time due to ecological filtering / selection. We set $\rho_1 = 0.01$ and $\rho_2 = 0.5$ to span a range comparable to connectivity values observed in speciation scenarios.

New species are introduced once community dynamics reach equilibrium, i.e., when the dynamics governed by Eq. (1) have

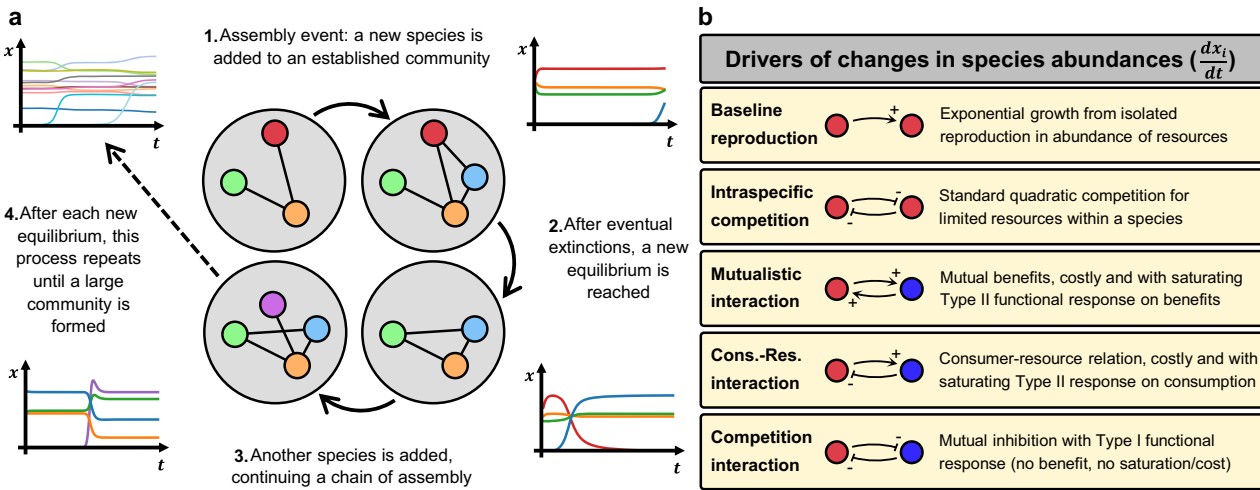

**Fig. 1 | Conceptual overview of the community assembly model and its ecological drivers. a** Communities assemble through the sequential introduction of new species, which may increase in abundance and may competitively exclude existing species, leading to extinctions and the establishment of a new ecological equilibrium. Despite such extinctions, species richness increases over time, resulting in the progressive build-up of a complex community. Alternative assembly scenarios differ in how ecological interactions are assigned between newly introduced and resident species. In the schematic, coloured circles represent species and edges denote the presence of ecological interactions. The plots illustrate the sequence of steps as species abundances ($x$), identified in the network by their colours, change in time ($t$). As an assembly event, the introduction of the blue species into a three- species community (step 1) triggers the extinction of the red species (step 2). Following the attainment of a new equilibrium, the purple species is introduced (step 3), and the assembly process is repeated until a large community is formed (step 4). **b** Species abundances are governed by standard ecological processes captured by a Lotka-Volterra framework, including logistic growth with baseline reproduction and intraspecific self-regulation due to competition for limited resources, as well as the three principal classes of interspecific interactions: mutualism, consumer-resource interactions, and competition. Costs to baseline reproduction and saturating functional responses are applied to positive interactions (mutualism and consumption), capturing both the maintenance costs and diminishing returns associated with accrued benefits.

stabilised. Species addition can destabilise the system, driving existing species below an extinction threshold and leading to their removal from the community. The addition of a new thriving species (i.e., that does not go extinct) constitutes an assembly event (Fig. 1a). After a new species is introduced in an assembly event, we assume no rewiring of its interactions, only the possible addition of interactions with new species included in subsequent events. Species abundances follow ecological dynamics governed by the system of differential equations in Eq. (1), where $x_i$ is the abundance of species $i$ in a community of $S$ species, governed by several mechanistic drivers (Fig. 1b). The non-negative coefficients $p_{ij}$, $m_{ij}$, and $c_{ij}$ represent the strengths of consumer-resource (+/-), mutualistic (+/+), and competitive (-/-) interactions, respectively, and are nonzero only when there is an interaction of the corresponding type between $i$ and $j$. These interactions are interspecific, thus $p_{ii} = m_{ii} = c_{ii} = 0$. These coefficients determine the strength of the effect of species $j$ on the density of species $i$. For consumer-resource interactions, we have $p_{ij}^+$ if $i$ is a consumer and $p_{ij}^-$ if $i$ is a resource.

$$\frac{1}{x_i}\frac{dx_i}{dt} = \underbrace{r_i - n_i^+ \delta}_{\text{baseline reproduction}} \underbrace{- s_i x_i}_{\text{intraspecific competition}} \underbrace{- \sum_{j=1}^{S} c_{ij} x_j}_{\text{competition interactions}} +$$

$$+ \sum_{j=1}^{S} \left( \underbrace{- \frac{p_{ij}^- x_j}{(1 + h_p \sum_k p_{jk}^+ x_k)}}_{\text{C-R interactions as resource}} + \underbrace{\frac{p_{ij}^+ x_j}{(1 + h_p \sum_k p_{ik}^+ x_k)}}_{\text{C-R interactions as consumer}} + \underbrace{\frac{m_{ij} x_j}{(1 + h_m \sum_k m_{ik} x_k)}}_{\text{mutualistic interactions}} \right)$$

(1)

The variable $n_i^+$ measures the number of interactions in which species $i$ is a mutualist or a consumer, used to consider the cost of every positive interaction. The sums over $j$ and $k$ run for all $S$ species. We used Type II functional responses to represent intake saturation in both mutualistic and consumer interactions. Note the subtle difference in the denominators of resource and consumer interaction terms, summing respectively over $p_{jk}^+$ and $p_{ik}^+$, because the sum gathers all the resources of a consumer (i.e., all sources). Each species $i$ has a positive intrinsic growth rate $r_i$ drawn from a half-normal distribution $|\mathcal{N}(\mu_r, \sigma_r^2)|$. Intraspecific competition strength $s_i$ is chosen such that the resulting carrying capacities $r_i/s_i$ follow a lognormal distribution. New interactions introduced via speciation or invasion are assigned a strength drawn from a half-normal distribution $|\mathcal{N}(0, \sigma^2)|$, where $\sigma$ sets the level of interaction strength (the choice of interaction type depends on the scenario). In the case of consumer-resource interactions, we impose a directional constraint on interaction strengths to reflect energetic inefficiency. Specifically, we assume that the number of resource individuals removed through consumption always exceeds the number of new consumer individuals produced as a consequence of that consumption. Accordingly, when interaction strengths are randomly sampled, if $p_{ij}^+ > p_{ji}^-$, we reset $p_{ij}^+$ so that $p_{ij}^+ = p_{ji}^-$. This constraint prevents unrealistically efficient trophic conversion.

To capture saturation of resource intake, we assign two parameters: $h_p$ for consumers and $h_m$ for mutualists, representing their average handling times. Every positive interaction (whether a consumer-resource link in which species $i$ is the consumer, or a mutualistic link in which $i$ benefits) incurs a harvesting cost $\delta$, representing the energetic and physiological investment required to acquire and maintain benefits from a given interaction. Such costs arise from the allocation of finite resources to interaction-related traits. For simplicity and interpretability, we assume a constant per-interaction cost, understood as a mean-field approximation capturing the average energetic burden of maintaining a positive interaction. This cost is subtracted from $r_i$ for each beneficial interaction, reflecting a trade-off: as a species accumulates positive links, it becomes increasingly reliant on its partners and may ultimately depend on them for survival (i.e., obligate mutualism or increasing trophic dependence on external resources for consumers, with reduced viability in their absence). This

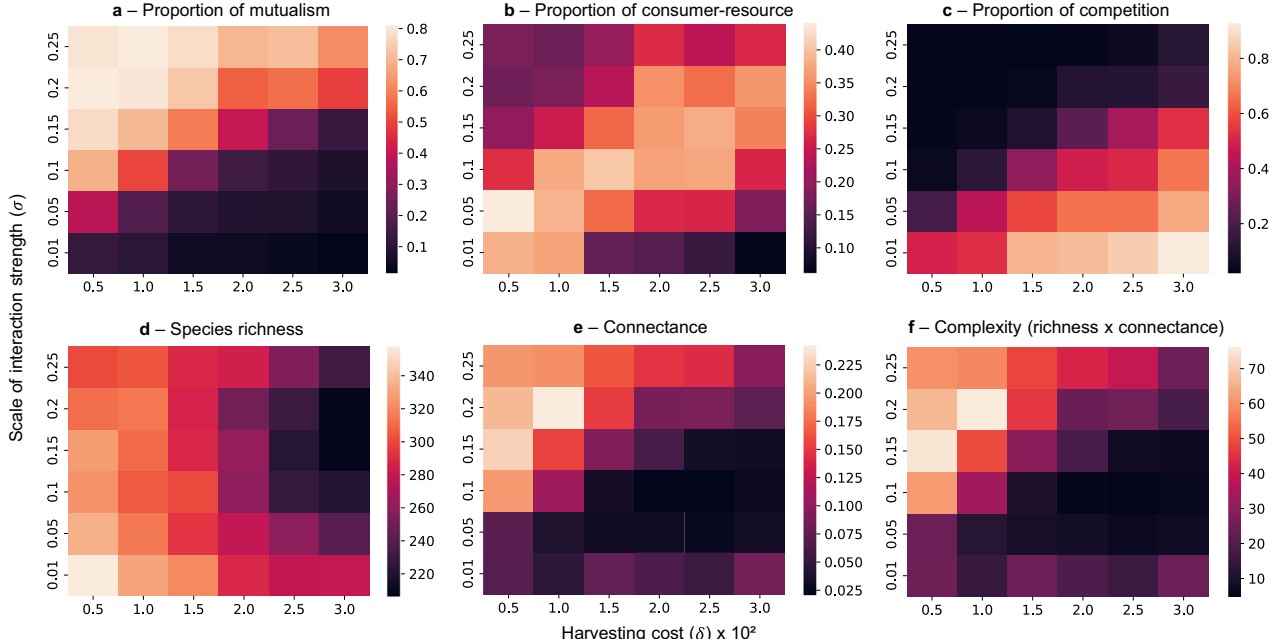

**Fig. 2 | The benefit of ecological interactions creates a threshold that separates two community types.** Final community structure after 500 assembly events under evolutionary assembly with strong inheritance (Δ = 5), shaped by the interplay between interaction strength (σ) and the cost of positive interactions or harvesting cost (δ). **a–c** The proportions of interaction types reveal a threshold separating two distinct community regimes. Type 1 communities, where interaction benefits are weak (low σ or high δ), are dominated by competition. Type 2 communities, where benefits are strong (high σ and low δ), are dominated by mutualism. Consumer-resource interactions have a higher proportion at the interface between the two regimes, particularly at low interaction costs.
**d–f** Community complexity, defined as species richness multiplied by connectance, is higher in Type 2 communities and follows the same pattern as connectance, as species richness only declines under high cost and high interaction strength. Values in heatmaps represent means across n=15 replicate simulations for each parameter combination. Interaction strengths were drawn from a half-normal $|\mathcal{N}(0, \sigma^2)|$. Communities started with 5 non-interacting species, and new species were introduced by speciation with Δ = 5 and initial abundances equal to the extinction threshold ($10^{-6}$). $h_p = h_m = 0.1$, $p_{ij}^- \geq p_{ji}^+$, $r_i \sim |\mathcal{N}(\mu_r, \sigma_r^2)|$.

choice reflects the allocation of energy in either interaction traits or survival and reproduction. The balance between the interaction strength parameter σ and the per-link cost δ therefore determines the net benefit of positive interactions.

## A separation between competitive and mutualistic communities

*The interplay between the strength and the cost of interactions creates a threshold that separates competitive from mutualistic communities.* After 500 evolutionary assembly steps, we observed a threshold in the benefit of ecological interactions that separates two distinct community types (Fig. 2). When σ is low and δ is high, resulting in weak interaction benefits, communities are dominated by competitive interactions (Fig. 2c). We refer to this regime as Type 1 (T1). Conversely, when σ is high and δ is low, producing strong benefits from ecological interactions, mutualisms predominate (Fig. 2a). We refer to this regime as Type 2 (T2). Consumer-resource interactions reach their highest proportion at the interface between T1 and T2, corresponding to intermediate levels of interaction benefit (Fig. 2b).

Species richness (i.e., the number of species in the network) decreases with increasing harvesting cost δ (Fig. 2d), while network connectance ($C = L/S^2$ on the binary directed network, a measure of connectivity) reflects the threshold between community types, being higher in T2 communities (Fig. 2e). As a result, community complexity, defined as the product of species richness and connectance, substantially increases in mutualism-dominated T2 communities (Fig. 2f). Additionally, we found that the intensity of intraspecific competition shifts the position of the threshold (Supplementary Fig. S2), with higher values shifting the regime from T2 to T1, indicating that the benefits of interactions must be evaluated relative to the strength of self-regulation. The existence of this threshold highlights interaction

benefits as key drivers of community assembly, shaping the composition of interaction types and mediating the emergence of complexity.

A simple mean-field argument can clarify the origin of the threshold. Consider a community structured by mutualistic and competitive interactions. The potential advantage of a species acquiring a new mutualistic link can be expressed by the inequality

$$\frac{\mu x}{1 + h_m k_1 \mu x} - \delta > 0, \tag{2}$$

where $x$ is the average abundance of a species, $\mu$ is the mean interaction strength, $k_1$ is the mean number of mutualistic links per species, and δ is the cost per mutualistic interaction. This inequality formalises the intuitive idea that a new mutualistic interaction is selectively favourable only if the associated benefit exceeds its cost. We can further approximate the equilibrium abundance of a species using a mean-field description:

$$\frac{1}{x}\frac{dx}{dt} = 0 = r - \delta k_1 + \frac{k_1 \mu x}{1 + k_1 h_m \mu x} - k_2 \mu x - \gamma x, \tag{3}$$

where $k_2$ represents the strength of interspecific competition and γ the mean intraspecific competition. Here, the focal species experiences the community as an average background, reducing the dynamics to a single uncoupled equation for $x$. By combining these two expressions, we can identify the parameter region in which the inequality is satisfied, given that $x > 0$ and $x$ satisfies the equilibrium equation. This analysis reveals the existence of a threshold in $k_1$, which shifts toward lower values depending on the parameter set.

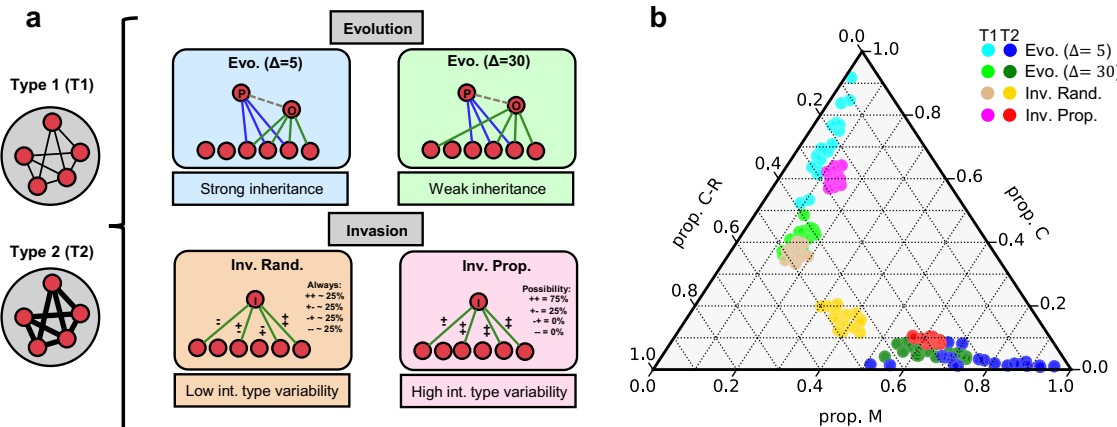

**Fig. 3 | Selection and inheritance of interactions drives the composition of interaction types in assembled communities. a** To disentangle the effects of inheritance from those of interaction-type selection, we analyse eight assembly scenarios, four for each community type (Type 1 and Type 2), spanning evolutionary and invasion processes. For evolution, we define strong ($\Delta = 5$) and weak ($\Delta = 30$) inheritance of interactions from parent to offspring. For invasion, we vary the level of interaction-type variability associated with the invader: low (random interaction types; Inv. Rand.) and high (random interaction-type proportions; Inv. Prop.). High interaction-type variability, present in both evolutionary scenarios and Inv. Prop., enables stronger interaction-type selection. Type 1 communities are defined by weak interaction benefits ($\sigma = 0.05$, $\delta = 0.025$), and Type 2 communities by strong benefits ($\sigma = 0.2$, $\delta = 0.01$). **b** Ternary plot showing the proportions of mutualistic (M), competitive (C), and consumer-resource (C-R) interactions across assembled communities. Each point represents a community at equilibrium after 500 assembly events, coloured by assembly scenario. Communities assembled under strong inheritance ($\Delta = 5$) show more extreme separation between community types, while those assembled via Inv. Rand. converge toward intermediate compositions due to limited interaction-type selection. Evolutionary scenarios produce more diverse interaction-type compositions across communities. Other parameter values are the same as in Fig. 2.

In particular, higher costs $\delta$ for a fixed small value of $k_2$ cause the inequality to fail at smaller $k_1$, highlighting a clear benefit-to-cost trade-off associated with acquiring additional mutualistic links (Supplementary Fig. S1). Intraspecific competition, being central in determining the equilibrium density $x$, also impacts the threshold.

## Selection and inheritance drive interaction type composition

Having established two distinct community types driven by interaction benefits, we next examine the role of inheritance in shaping community structure. To do so, it is necessary to distinguish inheritance effects from ecological filtering driven by the selection of interaction types that characterises the compositions of T1 and T2 communities. Both interaction-type selection and inheritance contribute to assembly under evolution, but only selection is present in assembly by invasion. By comparing evolutionary with invasion-driven assembly both considering the selection of interaction types, we can isolate the effects of inheritance.

To this end, we designed four assembly scenarios. These scenarios combine two community types (T1 and T2), two inheritance regimes (weak and strong, for evolution), and two levels of variability of interaction types (low and high, for invasion), yielding a total of eight scenarios (Fig. 3a). The key distinction among these scenarios lies in how new species acquire their ecological interactions during assembly events.

Evo. ($\Delta = 5$): evolution with strong inheritance. A new species arises via speciation from a randomly selected parent species. After copying the parent's interactions, up to five additions or deletions are allowed ($\Delta = 5$), resulting in high inheritance fidelity and close similarity between parent and offspring. The types of newly added interactions are determined by choosing the signs of each directed interaction (positive or negative) with equal probability.

Evo. ($\Delta = 30$): as above, but with $\Delta = 30$, offspring may differ by up to thirty interactions, producing low inheritance fidelity and greater divergence from the parent species. Intermediate evolutionary scenarios ($\Delta = 14$ and $\Delta = 22$) yielded a similar behaviour to this one (Supplementary Figs. S3 and S4, Supplementary Note 1).

Inv. Rand.: invasion with fixed probabilities for each interaction type. Each new species is introduced with independently assigned interactions, where each link is randomly assigned to one of the four available interaction types (mutualist, competitor, consumer, or resource). Due to the random nature of this process, this will yield a probability of 0.25 for each of the interaction types. Because these probabilities are fixed, the proportion of interaction types shows low variability across invaders, limiting the scope for ecological filtering (i.e., selection cannot select species with larger proportions of specific interaction types because they do not exist). This scenario serves as a baseline and mirrors previous models of invasion-based assembly with mixed interactions[18].

Inv. Prop.: invasion with variable proportions of interaction types. For each new invader, the overall proportions of the four interaction types are randomly drawn from a uniform distribution. Interactions are then assigned accordingly. This introduces high variability in the species-level composition of interaction types, allowing ecological dynamics to selectively shape (via selection) the structure of community-level interactions composition.

Community types (T1 and T2): for every scenario, we chose particular points ($\delta$, $\sigma$) to represent T1 and T2 community regions, ($\delta = 2.5 \times 10^{-2}$, $\sigma = 0.05$) for T1 (fifth column from the left and second row from the bottom in Fig. 2) and ($\delta = 1.0 \times 10^{-2}$, $\sigma = 0.2$) for T2 (second column from the left and fifth row from the bottom in Fig. 2). We tested different points within the same regions and all yielded similar analyses (Supplementary Figs. S5–S8, Supplementary Note 1).

After 500 assembly events and at ecological equilibrium, the eight scenarios produced a broad range of interaction-type compositions (Fig. 3b). Across all scenarios, the threshold between T1 and T2 communities was preserved: competitive interactions dominated in T1, while mutualistic interactions predominated in T2. However, inheritance and interaction-type selection each independently amplified the divergence between community types. Strong inheritance ($\Delta = 5$) led to the clearest predominance of the corresponding selected interaction type (consumer-resource or mutualistic), as well as the largest variability in the composition of interaction types. These results underscore the importance of species-level inheritance in shaping

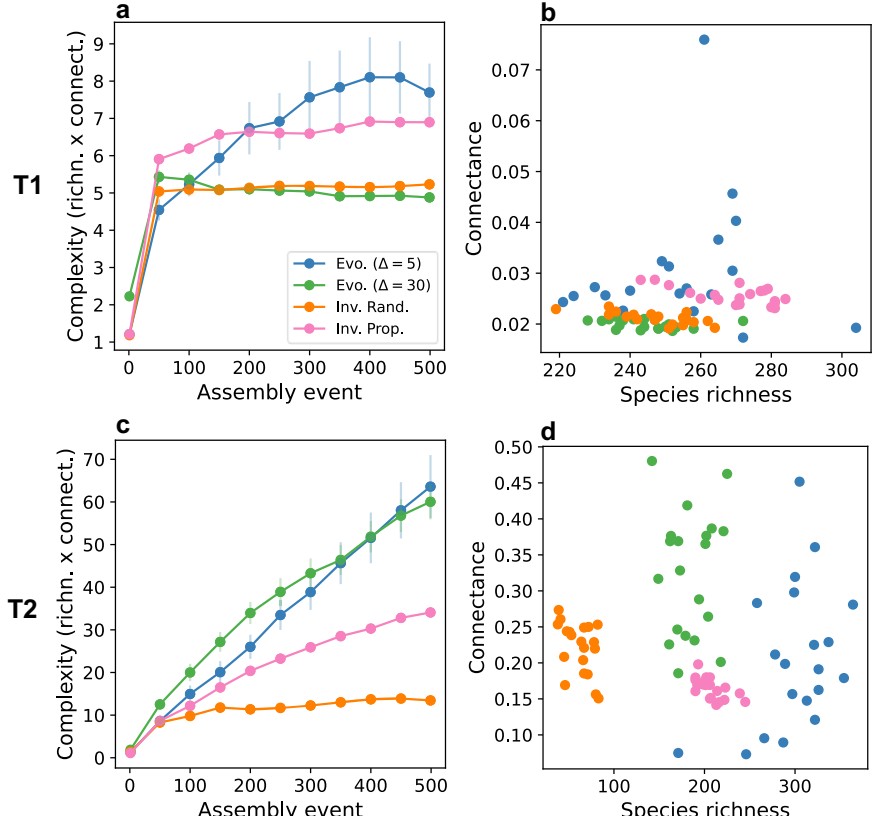

**Fig. 4 | Complexity emerges from selection and inheritance of interactions. a, c** Assembly histories showing the emergence of community complexity, defined as the product of species richness and network connectance, over 500 assembly events. Points represent the mean of n=20 replicate simulations sampled every 50 events, with vertical lines indicating the standard error ($SD/\sqrt{n}$). Type 2 communities consistently reach higher levels of complexity across all scenarios. In Type 1 communities, interaction-type selection increases complexity, while inheritance has minimal effect. In contrast, both selection and inheritance independently contribute to complexity in Type 2 communities. Weak and strong inheritance result in similar final complexity. **b, d** Final values of network connectance versus species richness for the communities shown in panels **a** and **c**, measured at assembly event 500. Each point represents a single simulation. Type 1 communities attain high species richness but low connectance, whereas Type 2 communities maintain higher connectance while increasing richness. Evolutionary scenarios produce greater variability in structural outcomes. Strong and weak inheritance yield high complexity through distinct routes, either via greater connectance or greater richness. Community types were specified as in Fig. 3, other parameter values are the same as in Fig. 2.

community-level structure and in reinforcing the effects of selection across community types.

## Evolution fosters diversity and complexity in assembled networks

To identify the drivers of community complexity, we recorded the trajectories of each assembly scenario across 500 assembly events, evaluated at ecological equilibrium (Fig. 4a, c). The increase in complexity ($S{\times}C$) was strongly dependent on community type, with T2 communities reaching substantially higher values across all scenarios. Across both community types however, both selection and inheritance independently contributed to increases in complexity. In T1 communities, Evo. ($\Delta = 30$) and Inv. Rand. exhibited similar trajectories, due to the low connectance inhibiting selection from effectively influencing the outcome of assembly when $\Delta$ is high. By contrast, Evo. ($\Delta = 5$) and Inv. Prop. communities reached higher average values of connectance and richness (Fig. 4b), leading to a more sustained increase in complexity following a rapid initial increase (Fig. 4a). Although species richness increased substantially in all communities, connectance remained consistently low. This suggests that when interaction benefits are low, resulting in competitive T1 communities, assembly tends to produce sparsely connected networks with limited complexity.

In T2 communities, the Inv. Rand. scenario produced communities with the lowest complexity, primarily due to reduced species richness (Fig. 4c, d). In contrast, Inv. Prop. resulted in higher complexity due to increased species richness, despite exhibiting lower average connectance (Fig. 4c, d). Evolutionary scenarios had a stronger impact on assembly outcomes than invasion. Both weak and strong inheritance produced similarly high levels of complexity, the highest among all scenarios (Fig. 4b), but through distinct structural pathways. Evo. ($\Delta = 5$) produced communities with higher richness and lower connectance, while Evo. ($\Delta = 30$) yielded the opposite pattern (Fig. 4d). Thus, the strength of inheritance acts as a mechanism shaping alternative routes to complexity. Moreover, evolutionary scenarios resulted in greater heterogeneity of communities across the richness vs. connectance space, compared to invasion (Fig. 4d). Overall, high interaction benefits that promote mutualism are also consistently associated with increased community complexity[26]. Both the selection of interaction types and inheritance contributed to the predominance of mutualism and the resulting emergence of complexity.

Our results suggest that the effectiveness of inheritance in driving the emergence of complexity depends on community type, and that the degree of inheritance determines the specific pathway through which complexity arises. T2 communities exhibit clearer distinctions between assembly scenarios, highlighting the stronger influence of both selection and inheritance in these mutualism-dominated systems.

We extended our analysis to explore faster evolutionary assembly events, out of equilibrium (Supplementary Figs. S9–S12,

Supplementary Note 1), mixed assembly events with different proportions of evolution and invasion (Supplementary Figs. S13 and S14), and varying levels of intrinsic growth rate (Supplementary Figs. S15–S18, Supplementary Note 1) and intraspecific competition (Supplementary Figs. S19–S22, Supplementary Note 1). All additional results contribute to corroborating the robustness of our conclusions. Moreover, these results show that evolution with high inheritance is more resilient than other scenarios in assembling a community that thrives under stricter conditions (Supplementary Figs. S15, S17, and S21).

To further examine how interaction architecture (i.e., the structure of who interacts with whom) emerges during community assembly, independently of interaction sign, direction, or magnitude, we analysed the topological degree entropy (a measure of the evenness in the number of links across species) and network modularity of assembled communities. For this analysis, community networks were represented in an undirected and unweighted form, retaining only the presence or absence of interactions. This choice isolates structural properties of interaction architecture that are robust to dynamical fluctuations in interaction strengths, while necessarily discarding information on interaction type, directionality, and energetic asymmetries. We quantified the relative increase in structure using a score metric of relative increase over random networks, comparing each observed value $Z$ to the average of 50 randomised communities $Z_r$ with matching complexity. The score was defined as $Z_s = (Z - Z_r)/Z_r$ (see Methods for details). In T1 communities, strong inheritance ($\Delta = 5$) produced a higher relative increase in both degree entropy and modularity. In T2 communities, both inheritance and selection contributed to increased entropy, but only inheritance led to a substantial increase in modularity (Supplementary Fig. S23). Weighted metrics yield equivalent results for T2 communities, but T1 communities no longer exhibit an increase in degree entropy (Supplementary Fig. S24). These findings suggest that inheritance promotes the emergence of more modular networks with heterogeneous distributions of the number of interactions across species, both organisational hallmarks of ecological networks[37].

**Assembled communities recapitulate microbiome organisation**

We compared our model results to empirical data from microbial communities in an attempt to identify the mechanisms, assembly processes, or community types that best approximate observed macroecological patterns in these ecosystems. We focused on patterns of relative abundance distributions found in both cross-sectional and time-series microbial community data using data from the human microbiome across different body sites (palm, gut, and mouth) and comprising 3,121 microbial types found in n = 1341 samples[31]. As shown elsewhere[31,35], these data exhibit consistent statistical patterns in the standardised log-mean abundance distribution (MAD) and in pairwise abundance correlations (Fig. 5). Importantly, deviations from random expectations in pairwise abundance correlations have been linked, in part, to ecological interactions[35]. To generate data from our model outputs comparable to those from microbial communities, we ran stochastic simulations of communities assembled from equilibrium after 500 assembly events (see Methods for details). To quantify the comparison between correlation distributions, we used the Wasserstein distance, which measures the total displacement of probability mass required to completely match two distributions.

We used the Inv. Rand. scenario, in which interactions are assigned randomly, as a baseline for comparison. Due to sparsity of interactions, all T1 scenarios produced similar relative abundance and pairwise correlation patterns (Supplementary Fig. S25, Supplementary Note 2), as illustrated in Fig. 5a, b for the Inv. Rand. baseline. This suggests that, in T1 communities, variation in assembly mechanisms does not translate into differences in abundance patterns, which may instead be dominated by environmental stochasticity. Moreover, none

of the T1 scenarios represented well the empirical MAD or pairwise correlation patterns.

In contrast to T1, T2 communities yielded a closer resemblance to the empirical patterns observed in microbial communities. The Inv. Rand. baseline scenario approximates the observable range of the microbial mean abundance distribution (MAD), though it fails to capture the upper tail of highly abundant taxa (Fig. 5c). It also reproduces the overall shape of pairwise abundance correlations (Fig. 5d). Weak evolutionary inheritance (Evo. $\Delta = 30$; Fig. 5e, f) and the Inv. Prop. scenario (Supplementary Fig. S26) yield an even closer match to empirical patterns. These scenarios better capture the higher density region of the MAD (Fig. 5e) and approximate the empirical distribution of negative pairwise correlations, although they underestimate the frequency of positive correlations (Fig. 5f).

Based on the Wasserstein distances (Supplementary Fig. S27) for the correlation pattern, T2 communities provide a better match to the empirical data than T1 communities. In T2 scenarios, the simulated distributions are closer to the empirical ones, and also closer to those obtained from the corresponding simulations without interactions (i.e. logistic simulations of the same communities). In contrast, T1 communities produce simulated distributions that deviate more strongly from the empirical data than their non-interacting counterparts, indicating that interactions in T1 communities systematically worsen the agreement with observations. Thus, whereas interactions in T2 communities improve empirical concordance, interactions in T1 communities drive simulations further away from empirical patterns. Extending this analysis to additional scenarios explored in the Supplementary Material yielded consistent results: good agreement was recovered for T2 simulations with mixed evolution-invasion assembly (Supplementary Fig. S28) and under reduced intraspecific competition (Supplementary Fig. S29), while no T1 scenario produced a comparably good match. Therefore, our analysis identifies T2 communities as better representations of microbial communities, highlighting strong interaction benefits as a potential mechanism underlying their macroecological abundance patterns.

## Discussion

The assembly of complex ecological communities is the result of evolutionary and ecological processes acting together to generate and maintain diversity. However, our understanding of how eco-evolutionary mechanisms interact to produce such diversity remains incomplete. Here, we investigated how interaction-type selection and inheritance shape the structure of species interaction networks in a dynamic community assembly framework. We identified two distinct community regimes defined by the level of benefit species derive from ecological interactions. When interactions are weak and costly, competitive links dominate. In contrast, stronger benefits promote mutualistic communities, with consumer-resource interactions increasing in proportion at the transition between these regimes. Evolution of interactions by speciation, implementing inheritance of interactions and interaction-type selection, was found to promote community complexity. The strength of inheritance modulates how complexity emerges: strong inheritance favours increased connectance, while weak inheritance promotes species richness. These results offer insights into the mechanisms underlying natural community assembly and reveal an association between interaction benefits and macroecological patterns observed in microbial communities.

Barbier and Loreau (2019)[38] demonstrated that food webs shift from stable, self-regulated 'pyramidal' structures to feedback-driven 'cascade' dynamics depending on the balance between interaction strength and intraspecific self-regulation. Similarly, our model identifies a threshold where the balance between species self-limitation (intraspecific competition) and how much they gain from others (benefits of interactions) determines whether mainly competitive (T1)

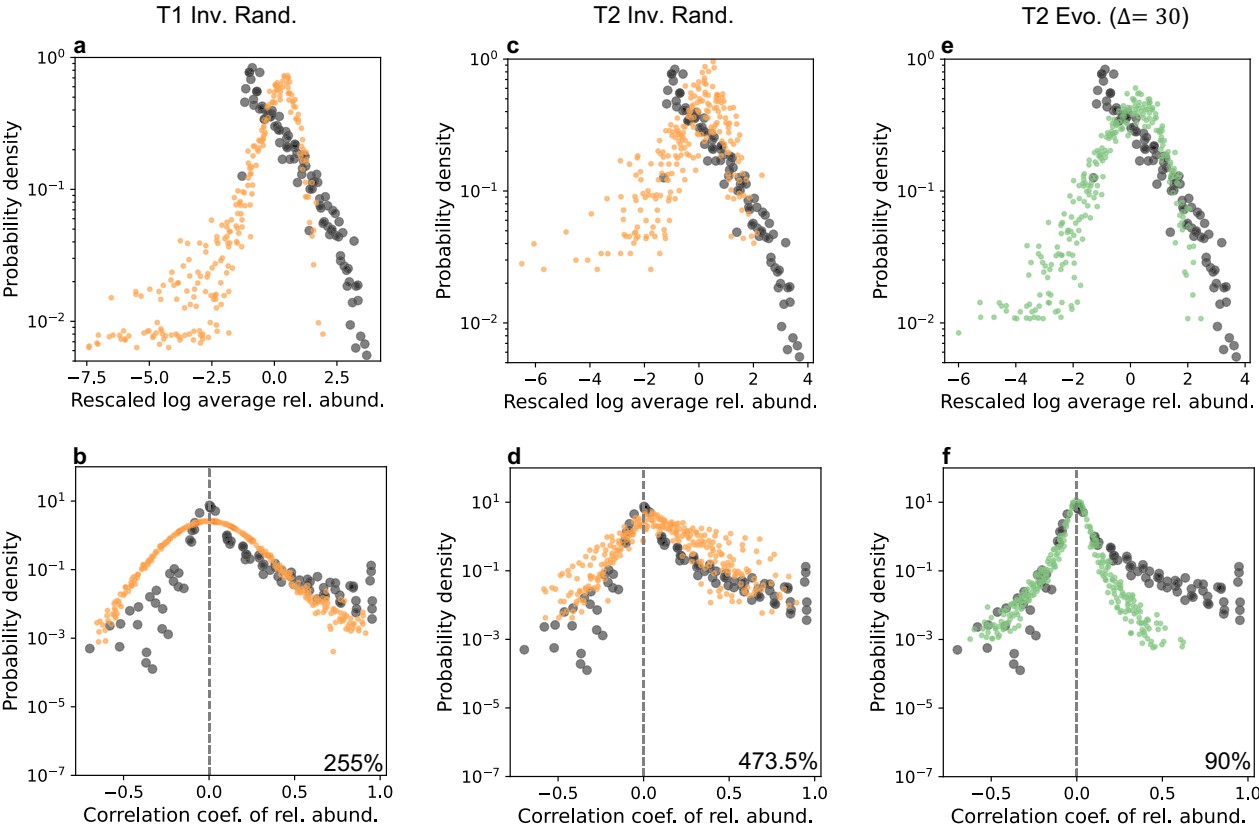

**Fig. 5 | Macroecological patterns observed in microbial communities are driven by ecological and evolutionary mechanisms.** We compared universal patterns from time-series data of human gut, palm, and mouth microbiomes (black points; see Methods) to simulated communities assembled via our eco-evolutionary model. The top row (**a, c, e**) shows the standardised log-mean relative abundance distributions (MAD), where each point represents the probability density of observing a species with a given abundance. Although real microbial abundances follow an approximately lognormal distribution, the lower tail is truncated due to sampling limitations. The bottom row (**b, d, f**) shows distributions of pairwise abundance correlations, where each point indicates the probability density of observing a given correlation coefficient between two species as their abundances fluctuate. Coloured points represent simulation outcomes. Percentages in panels **b**, **d**, and **f** quantify the relative difference between the average Wasserstein distance of simulated-empirical sample pairs and that of empirical-empirical sample pairs, defined as $(W_{se} - W_{ee})/W_{ee}$. $W_{se}$ denotes the mean simulated-empirical distance, and $W_{ee}$ denotes the mean empirical-empirical distance. A value of 100% indicates that

the typical distance between simulated and empirical samples is twice the typical distance among empirical samples. **a, b** Type 1 Inv. Rand. baseline scenario. All Type 1 scenarios yield distributions nearly identical to this baseline (Supplementary Fig. S25), and none approximate empirical MAD or correlation patterns. **c, d** Type 2 Inv. Rand. baseline: This scenario more closely matches the observed empirical distributions for both MAD and correlations, but fails to capture the high-abundance range in the MAD. **e, f** Type 2 weak inheritance (Evo. $\Delta = 30$): This produces a good fit to empirical data, capturing more of the observable MAD and reproducing the distribution of negative correlations, although positive correlations remain underestimated. Data sample sizes, $n = 131$ (feces F4), $n = 134$ (l-palm F4), $n = 334$ (feces M3), $n = 143$ (l-palm M3), $n = 268$ (r-palm M3), $n =$ and 331 (tongue M3). See Methods for full details. Simulations used the same parameters as in previous analyses, with $n = 20$ replicate communities sampled after 500 assembly events. Time-series fluctuations were generated by applying normally distributed environmental noise (s.d. = 0.1) proportional to species abundances, independently to each species.

or mutualistic (T2) networks emerge. Below the threshold, strong self-regulation or high interaction costs maintain communities in a competition-dominated regime. Above it, interaction benefits become sufficiently strong to favour the emergence of mutualistic networks. Consistent with our mean-field analysis, processes that broadly suppress species abundances, including stronger intraspecific competition, diminish the net gains from positive interactions and consequently favour the emergence of T1 communities. These findings support a general hypothesis: community structure is governed by the interplay between interaction strength, interaction costs, and self-regulation, which together drive the emergence of distinct and dynamically stable configurations.

The ability to manage energy and resources to sustain interactions has been proposed as a key driver of ecological diversity[39]. This supports the hypothesis that variation in the cost-benefit balance of interactions gives rise to distinct community types. Mooij et al. (2024)[40] showed that Lotka-Volterra competitive communities with sparse, weak interactions can stably support unlimited species

richness. These findings align with our results for Type 1 communities, which we propose emerge through assembly processes that favour competition over other interaction types. Previous studies have shown that both the mix of interaction types and their cost-benefit characteristics influence community stability and structure[14,16]. Our results demonstrate how these characteristics can be set to facilitate the emergence of ecological complexity, with high interaction benefits and the selection of mutualism appearing as important mechanisms, also in line with our previous explorations[26]. When speciation is included, these mechanisms lead to even higher complexity in Type 2 communities. These findings highlight the joint contribution of evolution and the structure of ecological interactions as drivers of community complexity. Surprisingly, high complexity emerged alongside stronger interactions, challenging May's classical view of random communities[11,12] and demonstrating that highly connected, strongly interactive systems can remain stable. Although May's inequality is true, we argue that a randomly defined community matrix (i.e., the Jacobian) does not

reflect complex communities with saturating positive interactions. Moreover, our results suggest that inheritance is a key driver of modularity, which is a property associated with stability in mutualistic and microbial communities[25,41,42].

Moreover, we also highlight the role of the ecological selection of interaction types as a generator of complexity, since it enhanced the emergence of complexity in Type 1 communities in the scenarios with effective selection (evolution with high inheritance and invasion with variable proportion of interaction types). This happened regardless of these scenarios featuring the highest proportions of competition, which means that complexity emerges through selection not only with a joint domination of mutualism. Both species richness and connectance were on average higher for these scenarios (Fig. 4b), although in Type 2 communities the effect of ecological selection of interaction types on complexity resulted from an increase in species richness only (Fig. 4d, difference between the two invasion scenarios).

In addition to promoting complexity and modularity, inheritance was essential for generating diverse community outcomes in interaction composition, connectance, and species richness. This variability highlights evolutionary processes (acting on species interactions) as key drivers of local and regional diversity, generating novelty in the co-occurrence patterns among communities. This conclusion aligns with the geographic mosaic theory of coevolution, which posits that local evolutionary dynamics generate population-level heterogeneity[1]. Our results show that local speciation, through species interactions alone, can drive divergent community outcomes, even in the absence of abiotic or trait-based heterogeneity. This further supports the role of priority effects and historical contingency in the evolutionary assembly of microbial communities[43,44]. Specifically, we find that the degree of inheritance shapes distinct pathways to complexity, favouring either species richness or network connectance. This richness-connectance trade-off has been observed in microbial systems[45], and we propose that inheritance strength acts as an evolutionary mechanism driving this pattern.

Differences between the Inv. Rand. and Inv. Prop. scenarios underscore the key role of interaction-type selection in shaping community assembly. However, successful invasions require an external supply of interaction-type diversity or a precise composition matching community demands, which may be rare or unreliable conditions. We argue that interaction-type selection is more characteristic of evolutionary assembly. Unlike invasion, evolution more reliably generates variability internally, through diversification and inheritance. Thus, even ecological filtering of interaction types may be primarily driven by evolutionary processes. Consequently, the influence of inheritance through speciation may be greater than suggested by the differences between the Inv. Prop. and speciation scenarios.

Camacho-Mateu et al. (2024)[35] found that sparse and weak interactions closely reproduce macroecological patterns observed in microbial communities. However, those configurations emerged from parameter optimisation rather than from a generative process based on internal community dynamics. Instead, we investigated the macro-ecological patterns exhibited by communities assembled via eco-evolutionary processes. While our results do not fully replicate empirical patterns, they reveal additional pathways through which these patterns may be approximated. Surprisingly, our results suggest that, when considering community assembly, strong and abundant interactions characterise microbial communities exhibiting such patterns. Moreover, previous studies linked phylogenetic distance with positive microbial abundance correlations[46]. We found that weak inheritance in T2 communities approximates empirical patterns well, except for positive correlations. This, combined with previous findings, suggests that modifying the implementation of weak inheritance, e.g., by altering how interactions evolve or incorporating taxon-specific traits, could improve alignment with empirical data. We also found that introducing species with randomly defined interactions can generate strong positive

correlations, offering an alternative explanation for observed pairwise patterns in microbial communities. Future work may build on these results to explore additional mechanisms that help explain the remaining variability in microbial macroecological patterns.

We highlight the contrast between inheritance through speciation and ecological filtering through selection of interaction types as distinct drivers of eco-evolutionary community assembly. Previous models incorporating mixed interaction types have examined invasion-driven assembly, similar to our Inv. Rand. scenario, and identified mutualism as a key factor in community persistence[18]. We extend these findings by isolating the contribution of interaction inheritance to community assembly. This distinction is an important step toward disentangling the relative contributions of ecological and evolutionary processes in community assembly. Disentangling these contributions is especially relevant for microbiomes, which are complex communities assembled in close phylogenetic and life-long association with host individuals[27,47]. These systems often reflect long-term coevolution among microbes and between microbes and their hosts[48], making evolutionary processes central to understanding microbiome structure and function. Future work could build on our framework to explore the mechanistic assembly of both free-living microbial communities and host-associated microbiomes to further characterise the action of evolutionary mechanisms. In this context, considering host communities alongside their associated microbes within a metacommunity framework, as recently proposed[27,49,50], may offer deeper insight into the mechanisms underlying the eco-evolutionary assembly of complex coevolved symbioses, highlighting the importance of dispersal and transmission modes.

Our results provide insights into the processes driving the emergence of complexity during eco-evolutionary community assembly. Key mechanisms include the strength of ecological benefits, the degree of inheritance of interactions, and the selection of interaction types, which determine whether competition or mutualism predominates. Understanding both the network structures and underlying mechanisms supporting stable, complex, and diverse communities is essential to explain their persistence in nature. This knowledge is also critical for predicting their responses to environmental change. In this way, our work advances ecological theory and supports efforts to maintain and restore complex ecological systems.

## Methods
### Simulations
All simulations begin with five species with no initial interactions. Species abundances at initialization are uniformly drawn from the interval [0, 0.02]. A species is considered extinct and removed from the system if its abundance falls below a threshold of $x_{ext} = 10^{-6}$ (we verified that results are qualitatively equivalent for $x_{ext} = 10^{-5}$ and $x_{ext} = 10^{-7}$ and thus not sensitive to a particular choice). Each simulation proceeds for 500 assembly events. Assembly occurs only after the system reaches ecological equilibrium, defined as a relative change in all species abundances of less than 0.01% over ten time steps of size 0.01. Intrinsic growth rates $r_i$ are sampled from a normal distribution $\mathcal{N}(\mu_r, \sigma_r^2)$ with $\mu_r = \sigma_r = 0.1$. Intraspecific competition coefficients $s_i$ are determined such that carrying capacities $K_i = r_i/s_i$ follow a lognormal distribution with underlying normal parameters $\mu = -2.2$, $\sigma = 0.5$; specifically, $s_i \propto r_i / \log \mathcal{N}(-2.2, 0.5^2)$. In evolutionary scenarios, $r_i$ and $s_i$ are not inherited (i.e. are sampled from their distributions, regardless of the parent), reflecting their dependence on species-specific environmental conditions and focusing inheritance on interaction structure only. Simulations use Type II functional response on positive interactions with constant handling times set as $h_m = h_p = 0.1$. When new interactions are introduced, their strengths are drawn from a half-normal distribution: $|\mathcal{N}(0, \sigma^2)|$, where $\sigma$ controls the typical strength of interactions. In evolution scenarios, inherited interaction strengths are subject to small stochastic variation, given by

$w' = w|(1 + \mathcal{N}(0, [0.05w]^2))|$, where $w$ is the parent's strength and $w'$ is the offspring's (interactions do not change type with inheritance). After the first 20 assembly events, all isolated species (i.e. those with no interactions) are removed after each subsequent event. Numerical integration of ecological dynamics (Eq (1)) was performed using the odeint function from the SciPy library in Python[51].

## Analysis

For each combination of $\delta = (\{0.005, 0.01, 0.015, 0.02, 0.025, 0.03\}$ and $\sigma = \{0.01, 0.05, 0.1, 0.15, 0.2, 0.25\})$ in the heatmaps, we ran 15 independent simulations using the strong inheritance evolution model ($\Delta = 5$), each consisting of 500 assembly events. Final community properties were averaged across these replicates and used to generate the heatmap data. To analyse the temporal dynamics of community composition (Fig. 4), we ran n=20 replicate simulations per scenario. Variables were recorded every 50 assembly events, and both the mean and standard error (defined as the standard deviation divided by the square root of the number of replicates) were computed and shown in the plots. Except for the heatmaps, all other results were based on these n=20 replicates per scenario. Type 1 and Type 2 communities were defined by parameter combinations ($\delta = 0.025$, $\sigma = 0.05$) and ($\delta = 0.01$, $\sigma = 0.2$), respectively (which are contained in the heatmaps).

## Network metrics

We evaluated two structural properties of the assembled ecological networks: degree entropy and modularity. Degree entropy quantifies the heterogeneity of the network's degree distribution $P(k)$, where $k$ denotes the number of interactions (degree) of a node (species). We computed the entropy using the unweighted network (i.e., considering only presence-absence of links), as $H = -\sum_k P(k)ln(P(k))$. Modularity is a standard network metric that quantifies the extent to which nodes form communities, i.e., groups of species that are more densely connected internally than to other parts of the network[52]. We calculated modularity from the unweighted interaction network using $Mod = \sum_c \left( \frac{L_c}{m} - \epsilon \left( \frac{k_c}{2m} \right)^2 \right)$ where $c$ indexes each community (or module), $L_c$ is the number of intra-community links, $k_c$ is the sum of degrees of nodes in community $c$, and $m$ is the total number of edges in the network. The resolution parameter $\epsilon$ was set to 1. All network analyses were performed using the Python package NetworkX[53], and community detection was carried out using the Louvain algorithm[54]. Effective increase: To facilitate comparison across scenarios, topological network metrics (degree entropy and modularity) were expressed as their effective increase relative to corresponding random networks. Specifically, for a given metric $Z$, we computed its effective increase $Z_s$ as $Z_s = \frac{Z - Z_r}{Z_r}$, where $Z$ is the observed value in the assembled network, and $Z_r$ is the mean value obtained from 50 Erdős-Rényi random networks with the same number of species $S$ and connectance $C$. This measure represents the percentage increase of the observed network property relative to what would be expected by chance. For the weighted comparison (Supplementary Fig. S24), we maintained information about interaction strength and summed the absolute strength ($m, p^+, p^-, c$) of the pair of directed interactions between two species to compose the weighted link.

## Data analysis

We utilised publicly available data from EBI Metagenomics: longitudinal human microbiome data (feces, left/right palm, tongue; subjects F4 and M3; Study MGYS00002184, ERP021896) from the EBI MGnify platform (project ERP021896)[55,56]. This is the same longitudinal dataset used by Grilli (2020)[31] to investigate macroecological patterns. We retained only samples containing at least 10,000 sequencing reads and computed species' relative abundances accordingly. The 6 filtered data groups utilised in the analysis contained n=131 (feces F4), n=134 (l-palm F4), n=334 (feces M3), n=143 (l-palm M3), n=268 (r-palm M3), and n=331 (tongue M3) samples. Two groups (tongue F4 and r-palm F4) were excluded due to their small resulting sample size. To generate comparable simulation data, we reused the same replicate simulations from Figs. 3 and 4, starting from the final equilibrium state reached after 500 assembly events. We modelled species abundance fluctuations using stochastic differential equations with environmental noise. Specifically, we added independent Gaussian noise to each species, scaled proportionally to its abundance, with a standard deviation of 0.1. Integration was performed using the Euler-Maruyama method, and abundances were recorded at evenly spaced, non-overlapping time intervals. We computed Wasserstein distances using the function wasserstein_distance from scipy.stats in Python. The simulations without interactions (logistic) used for comparison were done by integrating stochastic perturbations with only the intrinsic growth rates and intraspecific competition in the equation for the assembled communities. We waited until a new equilibrium was reached before sampling the timeseries.

## Reporting summary

Further information on research design is available in the Nature Portfolio Reporting Summary linked to this article.

## Data availability

No new experimental or field data were collected. Publicly available datasets from the EBI MGnify platform were used for model comparison, specifically longitudinal human microbiome data from subjects F4 and M3 in MGYS00002184. Raw OTU data were filtered and transformed as described in the Methods, and the original tables and process are available in the Zenodo repository [https://doi.org/10.5281/zenodo.15621027].

## Code availability

Files generated and analysed in this study and computer code developed to implement the model and execute the numerical simulations are available in the accompanying Zenodo repository [https://doi.org/10.5281/zenodo.15621027] and on GitHub: [https://github.com/computational-ecology-lab/eco-evo-network-assembly]. All simulations and data analyses were performed in Python (v3.11.11). The computational workflow relied on NumPy (v2.2.6) and SciPy (v1.15.2) for numerical computations, pandas (v2.2.3) for data handling, and Matplotlib (v3.9.1) and seaborn (v0.13.2) for visualisation. Additional specialised visualisations were generated using python-ternary (v1.0.8).

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

## Acknowledgements

This project was supported by the Leverhulme Trust through Research Project Grant # RPG-2022-114 to M.L. We acknowledge the support of the Supercomputing Wales project, which is part-funded by the European Regional Development Fund (ERDF) via Welsh Government.

## Author contributions

M.L. and G.A. conceived and designed the study. G.A. implemented the model and performed numerical simulations and analysis. M.L. and G.A. wrote the manuscript.

## Competing interests

The authors declare no competing interests.
