## [Transparent Peer Review file · Nature Communications]

The eco-evolutionary assembly of complex communities with multiple interaction types

Corresponding Author: Dr Miguel Lurgi

Version 0:

Reviewer comments:

Reviewer #1

(Remarks to the Author)

The manuscript is quite interesting and deals with a noteworthy topic in theoretical ecology. It shows two different types of communities emerging from assembly under different parameter regimes (interaction strength vs. harvesting cost), and then these two types are further investigated under different evolutionary inheritance (weak and strong) or variability of interaction types during invasions (low and high). This leads to 8 scenarios, for which complexity is examined in terms of species richness, connectance, and further network metrics. Relative abundance distributions are also compared to empirical observations on microbiomes.

I have few suggestions for clarifications and improvements, listed below. Further details are also attached.

Harvesting cost δ : this seems a relevant parameter in your model and investigation, but it is not very well explained or justified. Why is it needed? Is there any empirical or theoretical foundation for it? Why is it constant? The name itself is also not very informative.

New interactions for both evolutionary and invasion are extracted from a half-normal. The invasion scenarios Rand. and Prop. specify how the types of interactions are chosen, but it is unclear for the evolutionary scenarios.

Introduction of new species happens at equilibrium, are results robust if mutations and/or invasions were not rare?

Evolution and invasion are treated separately, what results would emerge if both were acting at the same time?

Among the network metrics, nestedness is not mentioned, but often used and discussed in this type of research.

Also, unweighted metrics are used, are results robust to the inclusion of weights in network metrics?

Comparisons with empirical data are qualitative, but the similarity between simulations and data should rather be quantified formally.

I hope these suggestions can help improve the manuscript.

(Remarks on code availability)

Reviewer #2

(Remarks to the Author)

The eco-evolutionary assembly of complex communities with multiple interaction types

In this study, the authors develop an eco-evolutionary community assembly model that integrates three interaction types (competition, consumer–resource, and mutualism) within a single dynamical framework. Communities assemble via sequential introduction of species, either through speciation with inheritance of interaction structure or via invasion from a regional pool. A cost–benefit trade-off is imposed on each positive interaction (mutualistic or consumer) via a per-link

harvesting cost and a saturating Type II functional response for the benefits. By exploring the plane of interaction strength (σ) and cost (δ), the authors identify a threshold separating two regimes: a competition-dominated, sparsely connected “Type 1” community and a mutualism-dominated, more connected “Type 2” community.

The paper then compares eight assembly scenarios that combine the two regimes (T1/T2), strong vs weak inheritance (Δ small vs large), and invasion with either fixed or variable interaction-type composition. This is used to separate the effects of selection of interaction types from inheritance of interactions on emergent complexity (richness \times connectance), degree entropy, and modularity. Finally, the authors compare time series generated from assembled communities to empirical macroecological patterns from human microbiome datasets, focusing on mean abundance distributions and pairwise abundance correlations.

Upon review, I am convinced that combining multiple interaction types with explicit inheritance in a community-assembly framework is timely and potentially impactful, especially given the connection to microbial macroecology. Generally, the model is well motivated and the results are interesting. However, I believe the manuscript can be substantially strengthened both conceptually and in clarity if the following major and minor points are addressed.

Major points

1. The dynamical system is central to the paper, yet Eq. (1) is currently difficult to parse. As typeset, the different sums and terms (competition, consumer–resource, mutualism, costs) are condensed in a way that is hard to reconstruct, and some notation (e.g. which sums appear in the denominators, the meaning of “ $\sum_{\{p+,m\}} \delta$ ”, etc.) is not fully explained.

I suggest:

- Rewriting Eq. (1) in a more modular way (e.g. grouping explicitly into “self-regulation”, “loss to consumers”, “gain from resources”, “gain from mutualists”, “harvesting cost” terms), possibly split over several lines.
- Explicitly defining all summation indices: over which sets (consumers, resources, mutualists) each sum runs, and which interactions contribute to the denominators of the Type II functional responses.
- Clarifying in the main text how consumer–resource asymmetry is handled (the rule $p_{\{ij\}}^+ \leq p_{\{ji\}}^-$ and the enforced symmetry when this fails). This is an important modelling choice and deserves a brief justification, not only a note in Methods.
- Stating explicitly whether the network is treated as directed and signed in the dynamics but undirected/unsigned when computing network metrics (see point 5).

2. The authors show a sharp transition in interaction-type composition, connectance, and “complexity” as σ and δ vary, and then select a single parameter pair in each region as “Type 1” and “Type 2”. Supplementary Fig. S1 further demonstrates that rescaling intraspecific competition shifts communities across this threshold, indicating that it reflects the balance of interaction benefits vs self-regulation, rather than σ and δ alone.

I suggest:

- Emphasising that T1 and T2 are regimes or “community types” defined by this relative balance, not single parameter points. The selected (σ , δ) pairs should be clearly described as representative.
- Providing at least a brief robustness check (in the SI) showing that key results for the assembly scenarios (e.g. the roles of inheritance and selection in Fig. 2–4) are qualitatively stable under modest changes in σ and δ within each regime.
- Considering a simple heuristic or mean-field argument for the location of the threshold (e.g. net per-capita growth at low density with typical numbers of positive links). Even a rough analytical back-of-the-envelope estimate would help readers see why a threshold should exist and how it depends on interaction benefits and self-regulation.

3. The contrast between strong vs weak inheritance is central to the eco-evolutionary framing, but the current description leaves some questions.

- The choice of $\Delta = 5$ vs $\Delta = 30$ feels somewhat ad hoc. While the qualitative distinction is clear, it would be helpful to comment on why these values were chosen and whether intermediate values (e.g. $\Delta = 10$ or 15) behave similarly to one of these extremes.
- Please clarify exactly how interaction changes are implemented when the parent has few links: is the number of deletions capped by the number of existing interactions? How is the balance of addition vs deletion chosen for a given d ?
- The model assumes inheritance acts only on interaction structure, not on intrinsic growth rates or self-regulation. This is a reasonable simplification, but it is worth stating explicitly in the Discussion that “evolution” here acts purely on interactions, which may limit the analogy to full trait or demographic evolution.

4. The Methods define an “effective increase” of degree entropy and modularity as $(Z - Z_r)/Z_r$, where Z_r is the mean for random networks. In the text, this is described as a “z-score metric”, but it is not a z-score in the usual statistical sense ($(Z - \text{mean})/SD$). This should be renamed to avoid confusion.

In addition:

- Both entropy and modularity are computed on the unweighted interaction network, seemingly ignoring sign and direction. Yet the underlying ecology is driven by signed, directed interactions (consumer vs resource, + vs –). Please state explicitly: – Whether the network is symmetrised and unsigned for these calculations.
- Why this choice is appropriate for the questions at hand, and what information might be lost.
- Since modularity in mutualistic or bipartite systems is often computed with more specialised methods, a brief acknowledgement of these simplifications would be useful, even if exploring signed or bipartite modularity is beyond the scope of this paper.

Minor comments

1. Fig. 1: The colour scales are clear, but please ensure that the palette is friendly to colour-blind readers. Also, it might help to mention in the caption that each heatmap cell is based on 15 replicate simulations.

2. Methods

- Explicitly state how connectance is computed (e.g. $C = L/S^2$ on the binary directed network).
- Mention that the extinction threshold (10^{-6}) has been checked and does not qualitatively affect results, if that is the case.
- As noted above, please rename the “z-score” to something like “relative increase over random networks” to align with the

formula.

In summary, this manuscript offers a valuable eco-evolutionary perspective on how multiple interaction types, inheritance, and ecological filtering jointly shape community structure and macroecological patterns. Addressing the points above, especially clarifying the dynamical model, sharpening the interpretation of the T1/T2 threshold, and strengthening the macroecological comparison, would substantially enhance its clarity, robustness, and impact.

(Remarks on code availability)

The code is clean, well organized, and easy to follow.

Reviewer #3

(Remarks to the Author)

This manuscript investigates the assembly of interaction networks, and how this process is shaped by inheritance and selection of interaction types. The study uses a Lotka-Volterra-type population dynamics model incorporating competitive, consumer-resource, and mutualistic interactions, and simulates scenarios in which new species are added either with or without inherited interactions. The main findings are: (1) the relative abundance of the three interaction types depends on the balance between interaction strength and interaction cost, (2) both inheritance and selection can promote network complexity, and (3) comparison with empirical microbial data suggests mechanisms that could have produced observed communities.

The novelty of this work lies in incorporating multiple interaction types, in contrast to most existing models that consider only one. This opens the door to addressing interesting questions in community assembly. The comparison with empirical data is also a strength. The manuscript is clearly written, the methodology is sound, and the figures are clear. However, several points would benefit from clarification, as I detail below.

Comment 1: overlap with previous study

There is substantial overlap with a previous publication (Araujo & Lurgi, 2025, "Mutualism provides a basis for biodiversity in eco-evolutionary community assembly". PLoS Comput Biol.), which not acknowledged.

First, both papers use the same eco-evolutionary model. While it is reasonable to apply the same model to explore different questions, the authors should acknowledge this explicitly (e.g., "We used a previously developed model..." in L119 rather than "We developed...").

Second, the published study focused on scenarios with high strength-to-cost ratios that yielded predominantly mutualistic (T2) communities. The present manuscript expands the parameter space to explore conditions generating predominantly competitive (T1) or consumer-resource communities. However, the results concerning network complexity in mutualism-dominated communities (L184–189) and the role of evolution (L258–272) appear to be essentially the same in both studies. Again, this is not acknowledged.

In summary, the manuscript should clearly outline similarities and differences with the previous study or reduce repetition where possible.

Comment 2: rationale for inheritance of interactions

The mechanism by which species "inherit interactions" from parents may seem abstract. One could view this as inheritance of traits that cause offspring to interact with similar partners as their parents. The mechanism itself is reasonable, but providing more biological justification or framing in the Introduction would help readers understand this modelling choice.

Comment 3: model details

The following clarifications would improve the model description:

(a) It may be helpful to describe in words what each term in Equation 1 represents (e.g., "The first term represents intrinsic growth..." etc.).

(b) My understanding is that interspecific interaction terms (competitive, mutualistic, consumer-resource) are nonzero only for interacting species pairs (i.e., c_{ij} , m_{ij} , $p_{ij} = 0$ for non-interacting pairs). Please clarify whether this is correct.

(c) It is unclear how initial interactions are defined. My understanding is: at the start there are five species with no interactions; under invasion, a new species is introduced and is connected to some existing species according to the drawn connectivity; under evolution, a new species inherits interactions from an existing species. If this is the case, how can interactions be inherited if the initial five species have none?

(d) My understanding is that there is no rewiring of interactions during ecological dynamics or after species addition. Species may lose interactions if their partners go extinct, which is the selection mechanism, but interactions are otherwise static. If correct, this simplification should be acknowledged, as rewiring is common in natural systems.

(e) Including a conceptual figure of the model and simulation workflow would be helpful.

Comment 4: effect of intraspecific competition

The result in L189-192 and Figure S1 is interesting, but discussed only briefly. For example, T1 communities arise when σ/δ is low and intraspecific competition (s) is low (as in the main text), but also when σ/δ is high and s is high. Could the authors speculate on why the magnitude of intraspecific competition reverses the effect of the strength-to-cost ratio?

Comment 5: complexity in T1 communities

The discussion focuses on the emergence of complexity in T2 communities (L341–343, L369-370). Could the authors also comment on complexity in T1 communities? In contrast to T2 communities, the Evo ($\delta=5$) and Inv.Prop. scenarios produce higher complexity than Evo ($\delta=30$) and Inv.Rand.

Comment 6: comparison with empirical data

This comparison is an interesting addition. However, simulation results do not appear highly sensitive to the eight scenarios simulated. Have the authors examined sensitivity to other model parameters, such as intrinsic growth rate or intraspecific competition (which seems to have substantial effects elsewhere)?

Specific comments

- L23: stability was not evaluated in this study and should be removed from this sentence
- L275-276: the phrase "exhibit more structured properties" is somewhat vague, please consider clarifying what is meant

(Remarks on code availability)

I have not reviewed the code as I am not familiar with its language - Python.
However, the README file is detailed and provides instructions for running the code.

Version 1:

Reviewer comments:

Reviewer #1

(Remarks to the Author)

The Authors carefully addressed all my concerns, and as far as I can tell also those from other Referees. I do not have any additional major considerations, however I attach a few minor suggestions on the main text and supplementary materials.

(Remarks on code availability)

I only accessed the zenodo link, which works, and codes and results are available, together with a readme file

Reviewer #2

(Remarks to the Author)

I am happy with the revised manuscript. The authors have carefully addressed all of my comments, and the revisions have clearly improved the clarity and presentation of the model, as well as the interpretation of the results.

Overall, the manuscript is now clear, well motivated, and ready for publication.

(Remarks on code availability)

Reviewer #3

(Remarks to the Author)

I thank the Authors for addressing my comments and answering my questions. I do not have any further comments. I believe this work to be an interesting contribution to the field.

(Remarks on code availability)

Reviewer #1 (Remarks to the Author)

The manuscript is quite interesting and deals with a noteworthy topic in theoretical ecology. It shows two different types of communities emerging from assembly under different parameter regimes (interaction strength vs. harvesting cost), and then these two types are further investigated under different evolutionary inheritance (weak and strong) or variability of interaction types during invasions (low and high). This leads to 8 scenarios, for which complexity is examined in terms of species richness, connectance, and further network metrics. Relative abundance distributions are also compared to empirical observations on microbiomes.

I have few suggestions for clarifications and improvements, listed below. Further details are also attached.

R. We thank the reviewer for the positive assessment of our manuscript and for the constructive suggestions. We have addressed all points raised and revised the manuscript accordingly. In particular, several of the reviewer's suggestions motivated additional simulations and robustness analyses, substantially expanding the scope of scenarios explored and strengthening the generality of our conclusions.

Harvesting cost δ : this seems a relevant parameter in your model and investigation, but it is not very well explained or justified. Why is it needed? Is there any empirical or theoretical foundation for it? Why is it constant? The name itself is also not very informative.

R. We agree that the rationale behind this parameter required clearer explanation. We have expanded the description of the harvesting cost δ in the main text (lines [188–200]) to provide a stronger biological motivation:

"[...] incurs a *harvesting cost* δ , representing the energetic and physiological investment required to acquire and maintain benefits from a given interaction. Such costs arise from the allocation of finite resources to interaction-related traits. For simplicity and interpretability, we assume a constant per-interaction cost, understood as a mean-field approximation capturing the average energetic burden of maintaining a positive interaction. This cost is subtracted from r_i for each beneficial interaction, reflecting a trade-off: as a species accumulates positive links, it becomes increasingly reliant on its partners and may ultimately depend on them for survival (i.e., obligate mutualism or increasing trophic dependence on external resources for consumers, with reduced viability in their absence). This choice reflects the allocation of energy in either interaction traits or survival and reproduction. The balance between the interaction strength parameter σ and the per-link cost δ therefore determines the net benefit of positive interactions."

New interactions for both evolutionary and invasion are extracted from a half-normal. The invasion scenarios Rand. and Prop. specify how the types of interactions are chosen, but it is unclear for the evolutionary scenarios.

R. Thank you for pointing this out. We have clarified this in the manuscript in lines [261-263], as part of the definition of evolution scenarios:

"The types of newly added interactions are determined by choosing the signs of each directed interaction (positive or negative) with equal probability."

Introduction of new species happens at equilibrium, are results robust if mutations and/or invasions were not rare?

R. We tested the robustness of our results to violations of the rare-event assumption by running additional simulations in which mutation (Evo 5) and invasion (Inv Prop) events occurred before ecological equilibrium was reached. These results are now presented in Supplementary Figures S9–S12. Overall, the qualitative patterns remained unchanged. Deviations were observed only in extremely fast assembly regimes, where limited relaxation time prevents communities from fully settling into their characteristic structural properties. We added this to our results in lines [336-337].

Evolution and invasion are treated separately, what results would emerge if both were acting at the same time?

R. We explored this by running additional simulations combining evolutionary (Evo 5) and invasion-based (Inv Prop) assembly events in varying proportions, now shown in Supplementary Figures S13 and S14. These simulations reveal gradual transitions between the patterns observed under pure evolution and pure invasion, with some nonlinear behaviour emerging when evolutionary events dominate. We highlight this in lines [336-342].

Among the network metrics, nestedness is not mentioned, but often used and discussed in this type of research.

R. We agree that nestedness is often discussed in the context of mutualistic communities. However, nestedness is typically defined for bipartite networks. Because our communities involve multiple interaction types and are not naturally bipartite, analysing nestedness would require additional assumptions about how to partition interactions. For this reason, we chose not to include nestedness metrics in the present study.

Also, unweighted metrics are used, are results robust to the inclusion of weights in network metrics?

R. We examined the robustness of our results to the inclusion of interaction weights, and present the corresponding analyses in Supplementary Figure S24. Most qualitative patterns remain unchanged. One notable difference is that weighted degree entropy in Type 1 communities shows a consistent decrease relative to random networks across all scenarios. Aside from this effect, the main conclusions are robust to the inclusion of weights. We report this in our results in lines [358-360].

Comparisons with empirical data are qualitative, but the similarity between simulations and data should rather be quantified formally.

R. We agree that a formal quantitative comparison strengthens the empirical relevance of the study. We have therefore extended the data analysis to include quantitative comparisons between simulations and microbial data, as well as control simulations without interactions between species. Specifically, we computed Wasserstein distances between probability distributions of abundance correlations and present these results in the new supplementary section “Complementary data analysis scenarios (Figs. S25–S29)”. This analysis confirms that Type 2 communities are generally more consistent with empirical macroecological patterns than Type 1 communities, a conclusion now explicitly stated in the revised main text (lines [400–416]).

I hope these suggestions can help improve the manuscript.

R. We thank the reviewer again for these helpful suggestions, which significantly improved the clarity, scope, and robustness of the manuscript.

Reviewer #2 (Remarks to the Author):

The eco-evolutionary assembly of complex communities with multiple interaction types. In this study, the authors develop an eco-evolutionary community assembly model that integrates three interaction types (competition, consumer–resource, and mutualism) within a single dynamical framework. Communities assemble via sequential introduction of species, either through speciation with inheritance of interaction structure or via invasion from a regional pool. A cost–benefit trade-off is imposed on each positive interaction (mutualistic or consumer) via a per-link harvesting cost and a saturating Type II functional response for the benefits. By exploring the plane of interaction strength (σ) and cost (δ), the authors identify a threshold separating two regimes: a competition-dominated, sparsely connected “Type 1” community and a mutualism-dominated, more connected “Type 2” community.

The paper then compares eight assembly scenarios that combine the two regimes (T1/T2), strong vs weak inheritance (Δ small vs large), and invasion with either fixed or variable interaction-type composition. This is used to separate the effects of selection of interaction types from inheritance of interactions on emergent complexity (richness \times connectance), degree entropy, and modularity. Finally, the authors compare time series generated from assembled communities to empirical macroecological patterns from human microbiome datasets, focusing on mean abundance distributions and pairwise abundance correlations.

Upon review, I am convinced that combining multiple interaction types with explicit inheritance in a community-assembly framework is timely and potentially impactful, especially given the connection to microbial macroecology. Generally, the model is well motivated and the results are interesting. However, I believe the manuscript can be substantially strengthened both conceptually and in clarity if the following major and minor points are addressed.

R. We thank the reviewer for the careful and thoughtful evaluation of our manuscript. We are pleased that the reviewer finds the framework timely and the results interesting. We have addressed all major and minor comments and believe that the revisions substantially improve the conceptual clarity, transparency of the model, and interpretability of the results.

Major points

1. The dynamical system is central to the paper, yet Eq. (1) is currently difficult to parse. As typeset, the different sums and terms (competition, consumer–resource, mutualism, costs) are condensed in a way that is hard to reconstruct, and some notation (e.g. which sums appear in the denominators, the meaning of “ $\sum_{\{p+,m\}} \delta$ ”, etc.) is not fully explained.

I suggest:

- Rewriting Eq. (1) in a more modular way (e.g. grouping explicitly into “self-regulation”, “loss to consumers”, “gain from resources”, “gain from mutualists”, “harvesting cost” terms), possibly split over several lines.
- Explicitly defining all summation indices: over which sets (consumers, resources, mutualists) each sum runs, and which interactions contribute to the denominators of the Type II functional responses.

R. We agree that the original presentation of Eq. (1) could be difficult to parse. We have therefore rewritten the equation in a more modular form and explicitly grouped terms according to their ecological roles (self-regulation, competitive effects, consumer-resource losses and gains, mutualistic gains, and intrinsic growth with costs). In addition, we now clearly define the scope of each summation and explicitly state which interaction terms contribute to the denominators of the Type II functional responses. These changes are described in the main

text (lines [166–173]) and are further supported by a new conceptual figure (Fig. 1B), which summarises all mechanistic components of the dynamical equation.

- Clarifying in the main text how consumer–resource asymmetry is handled (the rule $p_{ij}^+ \leq p_{ji}^-$ and the enforced symmetry when this fails). This is an important modelling choice and deserves a brief justification, not only a note in Methods.

R. We agree that this modelling choice deserves explicit justification in the main text. We have therefore expanded the explanation in lines [178–184]:

“In the case of consumer-resource interactions, we impose a directional constraint on interaction strengths to reflect energetic inefficiency. Specifically, we assume that the number of resource individuals removed through consumption always exceeds the number of new consumer individuals produced as a consequence of that consumption. Accordingly, when interaction strengths are randomly sampled, if $p_{ij}^+ > p_{ji}^-$, we reset p_{ij}^+ so that $p_{ij}^+ = p_{ji}^-$. This constraint prevents unrealistically efficient trophic conversion.”

- Stating explicitly whether the network is treated as directed and signed in the dynamics but undirected/unsigned when computing network metrics (see point 5).

R. We now explicitly clarify this point in the manuscript (lines [347–349]): “[...] we analysed the topological degree entropy (a measure of the evenness in the number of links across species) and network modularity in assembled communities. For this analysis, community networks were represented in an undirected and unweighted form, retaining only the presence or absence of interactions”

2. The authors show a sharp transition in interaction-type composition, connectance, and “complexity” as σ and δ vary, and then select a single parameter pair in each region as “Type 1” and “Type 2”. Supplementary Fig. S1 further demonstrates that rescaling intraspecific competition shifts communities across this threshold, indicating that it reflects the balance of interaction benefits vs self-regulation, rather than σ and δ alone.

I suggest:

- Emphasising that T1 and T2 are regimes or “community types” defined by this relative balance, not single parameter points. The selected (σ, δ) pairs should be clearly described as representative.

- Providing at least a brief robustness check (in the SI) showing that key results for the assembly scenarios (e.g. the roles of inheritance and selection in Fig. 2–4) are qualitatively stable under modest changes in σ and δ within each regime.

R. We agree that Type 1 and Type 2 communities should be interpreted as regimes defined by the balance between interaction benefits and costs, rather than as single parameter combinations. We have clarified this interpretation in the revised manuscript and explicitly describe the chosen (δ, σ) pairs as representative points within broader regions of parameter space (lines [284–289]):

“Community types (T1 and T2): for every scenario, we chose particular points (δ, σ) to represent T1 and T2 community regions, $(\delta = 2.5 * 10^{-2}, \sigma = 0.05)$ for T1 (fifth column from

the left and second row from the bottom in Fig 2) and ($\delta = 1.0 * 10^{-2}$, $\sigma = 0.2$) for T2 (second column from the left and fifth row from the bottom in Fig 2). We tested different points within the same regions and all yielded similar analyses (Figs S5-S8).”

- Considering a simple heuristic or mean-field argument for the location of the threshold (e.g. net per-capita growth at low density with typical numbers of positive links). Even a rough analytical back-of-the-envelope estimate would help readers see why a threshold should exist and how it depends on interaction benefits and self-regulation.

R. We thank the reviewer for this suggestion. We have added a simple mean-field analysis that considers the net per-capita gain of a focal species acquiring an additional positive interaction at equilibrium density. This analysis reveals a threshold in the space defined by the number of positive interactions and the per-link cost, beyond which accumulating additional positive interactions becomes detrimental. This threshold depends on self-regulation, which controls equilibrium abundances and therefore modulates the benefits of positive interactions. This analytical argument, presented in the main text (lines [224–243]) and Supplementary Figure S1, provides intuition for the emergence of the two community regimes observed in simulations.

3. The contrast between strong vs weak inheritance is central to the eco-evolutionary framing, but the current description leaves some questions.

- The choice of $\Delta = 5$ vs $\Delta = 30$ feels somewhat ad hoc. While the qualitative distinction is clear, it would be helpful to comment on why these values were chosen and whether intermediate values (e.g. $\Delta = 10$ or 15) behave similarly to one of these extremes.

R. We agree that the choice of Δ values requires justification. We therefore extended the analysis to include intermediate inheritance strengths ($\Delta = 14$ and 22), now presented in Supplementary Figures S3 and S4. These intermediate cases behave qualitatively closer to the weak-inheritance regime ($\Delta = 30$), indicating that our conclusions are not sensitive to the precise numerical values chosen for Δ in this case and that $\Delta=5$ successfully captures a strong-inheritance case. We present these findings in lines [266-267].

- Please clarify exactly how interaction changes are implemented when the parent has few links: is the number of deletions capped by the number of existing interactions? How is the balance of addition vs deletion chosen for a given d ?

R. We now clarified this in lines [136-140]: “For a given d , the number of interactions created is uniformly chosen from $[1, d]$ and the number of interactions destroyed is d minus that number. If there are not enough interactions inherited (or no interactions, as in the initial state), then new interactions are created whenever an inexistent one would be destroyed (and vice-versa).”

- The model assumes inheritance acts only on interaction structure, not on intrinsic growth rates or self-regulation. This is a reasonable simplification, but it is worth stating explicitly in the Discussion that “evolution” here acts purely on interactions, which may limit the analogy to full trait or demographic evolution.

R. We agree that this assumption should be stated explicitly. We now clarify in the Discussion that evolution is about interactions both in lines [427], “Evolution of interactions by speciation [...]”, and [483-484], “[...] evolutionary processes (acting on species interactions) [...]”.

4. The Methods define an “effective increase” of degree entropy and modularity as $(Z - Z_r)/Z_r$, where Z_r is the mean for random networks. In the text, this is described as a “z-score metric”, but it is not a z-score in the usual statistical sense $((Z - \text{mean})/SD)$. This should be renamed to avoid confusion.

R. We agree and have removed the labelling of a z-score metric, now stating “[...] a score metric of relative increase over random networks [...]”, in lines [352-353].

In addition:

- Both entropy and modularity are computed on the unweighted interaction network, seemingly ignoring sign and direction. Yet the underlying ecology is driven by signed, directed interactions (consumer vs resource, + vs –). Please state explicitly:

- Whether the network is symmetrised and unsigned for these calculations.

- Why this choice is appropriate for the questions at hand, and what information might be lost.

- Since modularity in mutualistic or bipartite systems is often computed with more specialised methods, a brief acknowledgement of these simplifications would be useful, even if exploring signed or bipartite modularity is beyond the scope of this paper.

R. We agree that this choice required clearer justification. We now explicitly state that degree entropy and modularity are computed on symmetrised, unsigned, and unweighted versions of the interaction networks. We clarify that this choice is deliberate, as our aim is to characterise the topological interaction architecture. We also explicitly acknowledge the information discarded by this simplification. These clarifications have been added in lines [343–351]:

“To further examine how interaction architecture (i.e. the structure of who interacts with whom) emerges during community assembly, independently of interaction sign, direction, or magnitude, we analysed the topological degree entropy (a measure of the evenness in the number of links across species) and network modularity of assembled communities. For this analysis, community networks were represented in an undirected and unweighted form, retaining only the presence or absence of interactions. This choice isolates structural properties of interaction architecture that are robust to dynamical fluctuations in interaction strengths, while necessarily discarding information on interaction type, directionality, and energetic asymmetries.”

Minor comments

1. Fig. 1: The colour scales are clear, but please ensure that the palette is friendly to colour-blind readers. Also, it might help to mention in the caption that each heatmap cell is based on 15 replicate simulations.

R. We confirm that all colour palettes used in the figures were selected to be colourblind-friendly. We have ensured consistency across figures. The figure caption (now Figure 2) specifies that the results presented are for 15 replicated simulations.

2. Methods

- Explicitly state how connectance is computed (e.g. $C = L/S^2$ on the binary directed network).

R. We now state it as suggested.

- Mention that the extinction threshold (10^{-6}) has been checked and does not qualitatively affect results, if that is the case.

R. We have tested and now explicitly stated that our particular choice of threshold does not qualitatively affect results, in lines [557-560]:

“A species is considered extinct and removed from the system if its abundance falls below a threshold of $x_{ext} = 10^{-6}$ (we verified that results are qualitatively equivalent for $x_{ext} = 10^{-5}$ and $x_{ext} = 10^{-7}$ and thus not sensitive to a particular choice).”

- As noted above, please rename the “z-score” to something like “relative increase over random networks” to align with the formula.

R. We renamed it as mentioned above.

In summary, this manuscript offers a valuable eco-evolutionary perspective on how multiple interaction types, inheritance, and ecological filtering jointly shape community structure and macroecological patterns. Addressing the points above, especially clarifying the dynamical model, sharpening the interpretation of the T1/T2 threshold, and strengthening the macroecological comparison, would substantially enhance its clarity, robustness, and impact.

R. We thank the reviewer for the insightful and constructive comments. We believe that the revisions have substantially improved the clarity, interpretability, and conceptual strength of the manuscript.

Reviewer #3 (Remarks to the Author):

This manuscript investigates the assembly of interaction networks, and how this process is shaped by inheritance and selection of interaction types. The study uses a Lotka-Volterra-type population dynamics model incorporating competitive, consumer-resource, and mutualistic interactions, and simulates scenarios in which new species are added either with or without inherited interactions. The main findings are: (1) the relative abundance of the three interaction types depends on the balance between interaction strength and interaction cost, (2) both inheritance and selection can promote network complexity, and (3) comparison with empirical microbial data suggests mechanisms that could have produced observed communities. The novelty of this work lies in incorporating multiple interaction types, in contrast to most existing models that consider only one. This opens the door to addressing interesting questions in community assembly. The comparison with empirical data is also a strength. The manuscript is clearly written, the methodology is sound, and the figures are clear. However, several points would benefit from clarification, as I detail below.

R. We thank the reviewer for the careful, thorough, and constructive evaluation of our manuscript. We particularly appreciate the attention given to issues of transparency, biological interpretation, and model clarity.

Comment 1: overlap with previous study

There is substantial overlap with a previous publication (Araujo & Lurgi, 2025, "Mutualism provides a basis for biodiversity in eco-evolutionary community assembly". PLoS Comput Biol.), which not acknowledged.

First, both papers use the same eco-evolutionary model. While it is reasonable to apply the same model to explore different questions, the authors should acknowledge this explicitly (e.g., "We used a previously developed model..." in L119 rather than "We developed...").

Second, the published study focused on scenarios with high strength-to-cost ratios that yielded predominantly mutualistic (T2) communities. The present manuscript expands the parameter space to explore conditions generating predominantly competitive (T1) or consumer-resource communities. However, the results concerning network complexity in mutualism-dominated communities (L184–189) and the role of evolution (L258–272) appear to be essentially the same in both studies. Again, this is not acknowledged.

In summary, the manuscript should clearly outline similarities and differences with the previous study or reduce repetition where possible.

R. We agree that this issue is important and thank the reviewer for raising it. The present manuscript builds on an eco-evolutionary modelling framework that we previously developed, and we now explicitly acknowledge this throughout the revised text. In particular, we have replaced phrasing suggesting model development with language clarifying that we *use a previously developed model*, and we cite the earlier study wherever appropriate. While both studies share the same modelling backbone, they address distinct research questions. The earlier paper focused on whether mutualism, under evolutionary assembly, promotes biodiversity, complexity, and stability, with an emphasis on mutualism-dominated communities. In contrast, the present manuscript investigates how the balance between interaction benefits and costs, together with inheritance and ecological selection of interaction types, gives rise to qualitatively distinct community regimes and determines their structural and macroecological properties. Some results for mutualism-dominated communities partially overlap between the

two studies. However, in the present manuscript these results are embedded in a different conceptual framework and are used to address different questions, particularly regarding regime emergence, the separation of inheritance from ecological filtering, and the consequences for community architecture. We have revised the manuscript to ensure that these connections are explicit and that any overlap is appropriately acknowledged.

Comment 2: rationale for inheritance of interactions

The mechanism by which species “inherit interactions” from parents may seem abstract. One could view this as inheritance of traits that cause offspring to interact with similar partners as their parents. The mechanism itself is reasonable, but providing more biological justification or framing in the Introduction would help readers understand this modelling choice.

R. We agree that this mechanism benefits from clearer biological framing. We included a better explanation as part of our definition of assembly by evolutionary speciation, before starting to present our results, in lines [126-129]:

“Inheritance with changes in species interactions represents mutations in foraging, morphological, or physiological traits that biologically determine interactions with other species (e.g. feeding modes, body morphology, metabolic pathways, signaling mechanisms, etc.).”

Comment 3: model details

The following clarifications would improve the model description:

(a) It may be helpful to describe in words what each term in Equation 1 represents (e.g., “The first term represents intrinsic growth...” etc.).

R. We agree, and we have now added a detailed verbal description of each term in Eq. (1) in the main text (lines [166–173]). In addition, we introduced a new conceptual figure (Fig. 1B) that summarises all mechanistic contributions to the ecological dynamics.

(b) My understanding is that interspecific interaction terms (competitive, mutualistic, consumer-resource) are nonzero only for interacting species pairs (i.e., c_{ij} , m_{ij} , p_{ij} = 0 for non-interacting pairs). Please clarify whether this is correct.

R. Yes, this is correct. We clarified this in lines [160-163]: “The non-negative coefficients p_{ij} , m_{ij} , and c_{ij} represent the strengths of consumer-resource (+/-), mutualistic (+/+), and competitive (-/-) interactions, respectively, and are nonzero only when there is an interaction of the corresponding type between i and j ”.

(c) It is unclear how initial interactions are defined. My understanding is: at the start there are five species with no interactions; under invasion, a new species is introduced and is connected to some existing species according to the drawn connectivity; under evolution, a new species inherits interactions from an existing species. If this is the case, how can interactions be inherited if the initial five species have none?

R. This point was also mentioned by reviewer #2, and this is our response as stated above: we now clarified this in lines [136-140]: “For a given d , the number of interactions created is

uniformly chosen from $[1, d]$ and the number of interactions destroyed is d minus that number. If there are not enough interactions inherited (or no interactions, as in the initial state), then new interactions are created whenever an inexistent one would be destroyed (and vice-versa).”

(d) My understanding is that there is no rewiring of interactions during ecological dynamics or after species addition. Species may lose interactions if their partners go extinct, which is the selection mechanism, but interactions are otherwise static. If correct, this simplification should be acknowledged, as rewiring is common in natural systems.

R. The reviewer is correct in this. We now acknowledge this simplification in lines [155-157]:

“After a new species is introduced in an assembly event, we assume no rewiring of its interactions, only the possible addition of interactions with new species included in subsequent events.”

(e) Including a conceptual figure of the model and simulation workflow would be helpful.

R. This is a very helpful suggestion and have added a new conceptual figure (Fig. 1) illustrating the community assembly process and the mechanistic components of the model. Figure 1A depicts the sequence of assembly events and ecological equilibria, while Figure 1B summarises the contributions of each term in the dynamical equation.

Comment 4: effect of intraspecific competition

The result in L189-192 and Figure S1 is interesting, but discussed only briefly. For example, T1 communities arise when σ/δ is low and intraspecific competition (s) is low (as in the main text), but also when σ/δ is high and s is high. Could the authors speculate on why the magnitude of intraspecific competition reverses the effect of the strength-to-cost ratio?

R. The reviewer is correct in their interpretation / assessment of the σ/δ effect, however a higher intraspecific competition favours T1 communities, while a lower intraspecific competition favours T2 communities. We now made our statement clearer on this aspect by adding in line [217-220]:

“we found that the intensity of intraspecific competition shifts the position of the threshold (Supplementary Fig S2), with higher values shifting the regime from T2 to T1, indicating that the benefits of interactions must be evaluated relative to the strength of self-regulation”.

We now also included a mean field exploration of the threshold (lines [224-243]). Our results indicate that the intraspecific competition changes the threshold by changing the mean species abundances, which in turn change how beneficial are the positive interactions.

Comment 5: complexity in T1 communities

The discussion focuses on the emergence of complexity in T2 communities (L341–343, L369-370). Could the authors also comment on complexity in T1 communities? In contrast to T2

communities, the Evo ($\Delta=5$) and Inv.Prop. scenarios produce higher complexity than Evo ($\Delta=30$) and Inv.Rand.

R. We have expanded the Discussion, now addressing complexity in Type 1 communities, in lines [471-480]:

“Moreover, we also highlight the role of the ecological selection of interaction types as a generator of complexity, since it enhanced the emergence of complexity in Type 1 communities in the scenarios with effective selection (evolution with high inheritance and invasion with variable proportion of interaction types). This happened regardless of these scenarios featuring the highest proportions of competition, which means that complexity emerges through selection not only with a joint domination of mutualism. Both species richness and connectance were on average higher for these scenarios (Fig 4B), although in Type 2 communities the effect of ecological selection of interaction types on complexity resulted from an increase in species richness only (Fig 4D, difference between the two invasion scenarios).”

Comment 6: comparison with empirical data

This comparison is an interesting addition. However, simulation results do not appear highly sensitive to the eight scenarios simulated. Have the authors examined sensitivity to other model parameters, such as intrinsic growth rate or intraspecific competition (which seems to have substantial effects elsewhere)?

R. We added a clarification in line [383] that results are not sensitive to scenarios in T1 communities due to the sparsity of interactions. We ran extra simulations varying the levels of intrinsic growth rate and intraspecific competition and presented results in Supplementary Figures S15-S22, and also mixed evolution and invasion scenarios presented in Figures S13 and S14. For the comparison with data, we presented the most interesting results among all these extra simulations in Figures S28 and S29, all for T2 communities. In addition to including some extra scenarios, we quantified the comparison with data, for the correlation pattern, using the Wasserstein distance between probability distributions, comparing data, simulations, and the logistic counterparts with no interactions between species. In this way, we could better assess the specific contribution of interactions in making the simulated pattern more similar to the data. The additional analyses made more robust our conclusion that T2 communities result in a better match to empirical patterns. However, it is hard to point out a better scenario overall, since there are many aspects to consider and the patterns were indeed similar across all scenarios, even considering the extra simulations. All these new analyses and quantitative analyses of empirical data matching are presented in lines [336-342] and lines [400-416].

Specific comments

- L23: stability was not evaluated in this study and should be removed from this sentence

R. We agree and have removed this statement.

- L275-276: the phrase “exhibit more structured properties” is somewhat vague, please consider clarifying what is meant

R. We agree that this phrasing was vague and have removed it.

Reviewer #1

The Authors carefully addressed all my concerns, and as far as I can tell also those from other Referees. I do not have any additional major considerations, however I attach a few minor suggestions on the main text and supplementary materials.

I only accessed the zenodo link, which works, and codes and results are available, together with a readme file

R. We thank the reviewer for their careful evaluation of the revised manuscript and for confirming that the main concerns have been satisfactorily addressed. We appreciate their additional minor suggestions and have incorporated them to the revised version of the manuscript. We thus have improved the conceptual figure, the presentation of equations, and included the Wasserstein distances in Fig 5.

Reviewer #2

I am happy with the revised manuscript. The authors have carefully addressed all of my comments, and the revisions have clearly improved the clarity and presentation of the model, as well as the interpretation of the results. Overall, the manuscript is now clear, well motivated, and ready for publication.

R. We thank the reviewer for their constructive feedback throughout the review process. We are pleased that the revisions have improved the clarity of the model and the interpretation of the results, and we appreciate the reviewer's positive evaluation of the final version.

Reviewer #3

I thank the Authors for addressing my comments and answering my questions. I do not have any further comments. I believe this work to be an interesting contribution to the field.

R. We thank the reviewer for their thoughtful comments and questions, which helped us strengthen the manuscript. We appreciate their positive assessment and their recognition of the contribution of this work to the field.